# Biological and Pharmacological Effects of Synthetic Saponins

**DOI:** 10.3390/molecules25214974

**Published:** 2020-10-27

**Authors:** Yu-Pu Juang, Pi-Hui Liang

**Affiliations:** School of Pharmacy, College of Medicine, National Taiwan University, Taipei 100, Taiwan; a98754687@gmail.com

**Keywords:** synthetic saponins, triterpenoid saponins, steroid saponins, bioactivity, mechanism

## Abstract

Saponins are amphiphilic molecules consisting of carbohydrate and either triterpenoid or steroid aglycone moieties and are noted for their multiple biological activities—Fungicidal, antimicrobial, antiviral, anti-inflammatory, anticancer, antioxidant and immunomodulatory effects have all been observed. Saponins from natural sources have long been used in herbal and traditional medicines; however, the isolation of complexed saponins from nature is difficult and laborious, due to the scarce amount and structure heterogeneity. Chemical synthesis is considered a powerful tool to expand the structural diversity of saponin, leading to the discovery of promising compounds. This review focuses on recent developments in the structure optimization and biological evaluation of synthetic triterpenoid and steroid saponin derivatives. By summarizing the structure–activity relationship (SAR) results, we hope to provide the direction for future development of saponin-based bioactive compounds.

## 1. Introduction

The name “saponin” is derived from the Latin word *sapo*, meaning soap-like foam-generating ability, and the amphiphilic properties derived from the structure containing an isoprenoid-derived aglycone (a sapogenin) attached to one or more sugar chains by either an ether or ester linkage. Structural classification of saponins is primarily based on their sapogenin skeletons, which can be divided into two main groups—Triterpenoid saponins and steroid saponins [1,2]. Triterpenoid saponins are broadly distributed in dicotyledons, including four major skeletons—Pentacyclic oleanane, ursane, lupane, and tetracyclic dammarane (Figure 1a); steroid saponins are mostly derived from monocotyledons, comprised of four major skeletons—Tetracyclic cholestane, hexacyclic spirostane, pentacyclic furostane, and lactone-bearing cardenolide (Figure 1b). Sugar-bearing sapogenins are categorized by numbers of sugar residue in monodesmosidic (one sugar residue), bidesmosidic (two sugar residues), and polydesmosidic saponins (three or more sugar residues) [3].

In nature, saponins are found in plants and marine animals, where they are implicated in host defense against pathogens and herbivores [4]. Since saponins are presented in many medicinal plants and Chinese herbal medicines, exhibiting a plethora of biological activities, including antifungal, antimicrobial, antiviral, anti-inflammatory, anticancer, antioxidant, and immunomodulatory effects, they can serve as a good starting point for the development of natural product-derived drugs [5]. However, the mechanism and SAR of saponins is poorly understood, and the isolation from plants to get appropriate amounts is sometimes troublesome and laborious due to microheterogeneity and scarcity of the molecules. As a result, applying an organic synthesis method to generate artificial saponins is a promising way to efficiently expand the structure library and search for highly active compounds [6,7].

The number of research articles related to saponin have risen sharply in the past decade, indicating that the field is receiving more and more attention, requiring the need to broaden structure diversity efficiently. The synthetic methods of saponins have been well reviewed, providing complete information on the organic synthesis to perform SAR optimization [6,7]. Based on the research experiences on carbohydrate [8,9] and medicinal [10,11,12,13,14,15,16,17,18,19,20,21,22] chemistry in our lab, we summarize the biological effects and preliminary SAR results of synthetic saponins over the last five years to serve as a bridge between chemistry and biology, providing an insight for further structure optimization and mechanism studies, finally facilitating the development of saponin-based bioactive compounds.

## 2. Triterpenoid Saponin

### 2.1. Oleanane

Oleanane-type saponins are the most studied synthetic saponins, due to their promising pharmacological effects and high natural abundance. Common oleanane-type skeletons modified with a chemical approach are oleanolic acid, hederagenin, and quillaic acid. They have been isolated from enormous plant species as either a free triterpenoid or a saponin and are particularly rich in the Oleaceae family [23]. Oleanane-type saponins are reported to exhibit multiple biological activities, especially antitumor, antiviral and immunomodulatory effects [24,25,26]. However, toxicity triggered by hemolytic and membrane lysis effect is the major challenge in drug development [27], and the understanding of structure–toxicity relationships is at an early stage [28].

#### 2.1.1. Oleanolic Acid

Synthetic oleanolic saponins (Figure 2) have been evaluated on their antitumor [14,18,21,29,30,31,32,33,34,35,36,37], anti-Alzheimer [38], antiviral [39,40], antiglucosidase [41], immunomodulating [42], and detoxification [43] activities (Table 1). Yang and co-workers reported a series of structures derivatized from potent antitumor *Pulsatilla* saponins, hederacolchiside A and D, elucidating the selectivity between the antitumor activity and hemolytic toxicity of compounds. The modification at 28-COOH of hederacolchiside was found to reduce toxicity with conservative antitumor efficacy, compared to hederacolchiside A_1_ (HA_1_). Compound **1** with glycine-derived amide [36] and **2** with methyl ester [33] showed a promising cytotoxic IC_50_ as 0.9–3.2 µM and 1.1–4.6 µM, respectively, against several human cancer cell lines and did not trigger acute toxicity in mice. With further modifications, ester-derived nitric oxide-donating compound **3** shows a cytotoxic IC_50_ in vitro between 1.6 and 6.5 µM and better in vivo antitumor efficacy than HA_1_ without acute toxicity [32]. The attempt to enhance antitumor activity by altering the sugar contents of the trisaccharide moiety on HA_1_ was unable to generate a better compound, indicating the construction of a hydrophilic sugar chain was not directly related to antitumor activity [34,44]. Cheng and co-workers constructed a series of HA_1_ derivative via Cu(I)-catalyzed azide-alkyne cycloaddition, and compounds modified with more lipophilic moiety (**4**) were found to exhibit higher antitumor activity than HA_1_ in a 0.5–2.7 µM range [35]. Further mechanistic studies revealed that the compound **4** could activate HepG2 cell apoptosis through mitochondria-mediated intrinsic apoptosis pathways.

Cheng and co-workers synthesized a series of albiziabioside A (AlbA) derivatives, of which the disaccharide analogues were less cytotoxic against cancer cells than parent compounds [37]. Therefore, the direction of structure optimization was changed to the modification of secondary amine residue on glucosamine, and the amide derivatives were found to be inactive against most of cancer cell lines except HCT116 colorectal cancer cells. Compound **5** exhibited cytotoxic selectivity between normal cells (>50 µM) and HCT116 (7.6 µM), and the antiproliferative activity might be related to mitochondria-dependent apoptosis [31]. To further improve antitumor efficacy, a secondary amine moiety of AlbA was conjugated with dichloroacetate (**6**), a pyruvate dehydrogenase kinase inhibitor, to disrupt glycolysis pathway in cells, leading to increased intracellular reactive oxygen species (ROS) and decreased accumulation of lactic acid in the tumor microenvironment. The cancer cell death was induced by caspase-dependent pathway activation, GPX4 pathway suppression, and lipid peroxidation accumulation, which can be summarized into apoptosis-ferroptosis-M2-TAM polarization [30]. Sun and co-workers constructed the synthesis method of four naturally occurring oleanolic saponins, revealing that monodesmosidic saponin exhibited higher antitumor cytotoxicity than bidesmosidic saponin [29].

Since 2014, our group had studied the *N*-acetyl glucosamine-bearing oleanolic saponins, which were found in plants and showed promising antitumor cytotoxicity against several cancer cell lines [44]. To investigate the influence of activity on the sugar chain, we constructed a series of oleanolic saponins with (1 → 3)-linked, (1 → 4)-linked, and (1 → 6)-linked *N*-acetylglucosamine oligosaccharide residues; it was found that derivatives bearing modifications to the 3′-glycosyl moiety and/or those incorporating _D/L_-xylose and _L_-arabinose were more cytotoxic to HL-60 and HT-29 cancer cell lines, but still less cytotoxic than the oleanolic saponin bearing a single *N*-acetylglucosamine (**7**) [14]. Based on this finding, we modified the secondary amine moiety and generated *N*-acyl, *N*-alkoxycarbonyl and *N*-alkylcarbamoyl derivatives with different lengths of carbon chains. Their cytotoxicity against HL-60 cell were related to the length of the carbon chain in a bell-shaped manner, with seven to nine carbons showed promising efficacy and the carbamate bridge being better than amide and urea (**8** and **9**, 0.76 and 1.40 µM respectively). Cell imaging revealed the cytosolic distribution of the compounds and the mitochondrial membrane potential loss identified with JC-1 dye made us believe that the cytotoxicity of saponins originated from the mitochondria-dependent pathway [18]. Interestingly, with flow cytometry analysis and detailed confocal imaging, the bell-shaped activity phenomenon was due to the cell penetrating effect of our compounds. Compounds with long carbon chains would stick on cell membranes, leading to lower cytotoxicity against cancer cells. To further elucidate the mechanism of our compound in causing HL-60 cell death, we applied the Annexin V/DRAQ5 assay to identify the apoptosis-triggering effect, and the cell cycle analysis found that HL-60 cancer cells were arrested in the sub-G1 phase in a dose-dependent manner. Combining the results of western blotting and stable isotope labeling by/with amino acids in cell culture (SILAC) proteomics, glucosamine-bearing oleanolic saponin was found to trigger both intrinsic and extrinsic apoptosis and inhibit the PI3K/Akt/mTOR pathway in HL-60 cancer cells [21].

Evaluation of the neuroprotective effect of hederacolchiside E derivatives bearing various sugar chains was performed by Cheng and co-workers; compound **10** was found to prevent cell death by reducing the release of lactate dehydrogenase, level of intracellular ROS, and malondialdehyde resulting from Aβ_1–42_ treatment in PC12 cells [38]. Song and co-workers generated a series of 3-*O*-β-chacotriosyl oleanolic saponins serving as entry inhibitors of H5N1, revealing that a disubstituted amide modification at 28-COOH (**11**), alteration of C-3 configuration from β to α (**12**), and introduction of an oxo group to C-11 of oleanolic acid (**13**) can enhance the selectivity index between antiviral and toxicity [39,40]. Kobayashi and co-workers explored the mucosal adjuvant activity of oleanolic saponins with a cinnamoyl-modified glycosyl moiety at 28-COOH for generating nasal anti-influenza virus antibodies, showing mild IgG and IgE enhancements [42]. They also revealed the ability of oleanolic acid 3-glucoside to suppress the methylmercury accumulation in vitro and in vivo, serving as a toxicity alleviation agent for methylmercury [43]. Liu and co-workers synthesized two derivatives based on a *Gypsophila oldhamiana* saponin, where they demonstrated that the C-4 OHs have better α-glucosidase inhibition activity than C-4 CHO or C-4 CH_2_OH [41].

#### 2.1.2. Hederagenin

Synthetic triterpenoid saponins based on hederagenin (Figure 3) have been reported to exhibit cytotoxicity against cancer cells [45,73,74] and antileishmanial activities [46] (Table 1). Yang and co-workers synthesized a series of hederagenin-type saponin derivatives from pulsatilla saponins A and D (PSA, PSD), bearing modifications to the sapogenin moiety. However, none of the saponin derivatives tested showed superior cytotoxicity in cancer cells compared to parent saponins and hederagenin derivatives [73,74]. To further expand the SAR of PSD, several 28-COOH esters and amide derivatives were synthesized, and compounds **14** and **15** showed better cytotoxicity in five human cancer cell lines (IC_50_ = 1.2–4.7 µM and 1.7–4.5 µM, respectively) and lower acute toxicity to mice than PSD. Moreover, compound **15** was found to kill HCT-116 through apoptosis, causing G1 cell cycle arrest, and showed similar antitumor efficacy in vivo compared to PSD [45]. A large library of semi-synthetic hederagenins and their saponins were evaluated on their antileishmanial properties, but could not acquire ideal selectivity index in eliminating axenic *L. Mexicana* amastigotes and toxicity to host a macrophage (RAW 267.4) [46].

#### 2.1.3. Quillaic Acid

Synthetic quillaic saponins (Figure 3), based on *Quillaja saponaria* saponins, are mostly studied for their immunomodulatory activities (Table 1) [75]. The first synthesis of *Quillaja* saponins fraction-21 (QS-21) was established by Gin and co-workers [76] and they accomplished extensive SAR studies, which have been completely summarized in a previous review [77]. Recently, Wang and co-workers dedicated themselves to the expansion of SAR knowledge in various isolating fractions of *Quillaja* saponins. In QS-21-based saponins, the derivatization on glucuronic acid was found to improve adjuvanticity with terminal carboxylic acid (**16**) or monosaccharide modification [47,78]. The derivatization of QS-17/18-based saponins with amide coupling could only achieve similar adjuvanticity to GPI-0100, a QS-21 derivative with a lower toxicity [79]. To investigate the relationship between the hexasaccharide on 28-COOH of QS-7 and immunomodulatory efficacy, a series of compounds bearing different sugar chains were synthesized, and compound **17**, with acetyl-modified pentasaccharide, was found to exhibit the most potent adjuvanticity, even similar to QS-21 [49,80]. Since the amounts of *Quillaja* saponin were extremely low in the bark of *Quillaja saponaria*, *Momordica* saponins were found to be a surrogate source of *Quillaja* saponins, and the amide-modified compound **18** can trigger better adjuvanticity than GPI-0100 [48,81]. The group of Fernandez-Tejada synthesized several 3-O-truncated QS-21 derivatives, revealing that the replacement of ester linkage by thioester at 28-COOH preserved the activity (**19**), and echinocystic acid was found to exhibit similar adjuvanticity to their quillaic congeners [82].

### 2.2. Ursane

Ursolic acid is a pentacyclic triterpenoid mostly isolated from berries, especially cranberries (*Vaccinium macrocarpon*), and other *Vaccinium* species [83]. Multiple biological activities are found in the synthetic ursolic acid derivatives (Figure 4), such as antiviral, cardioprotective, anticancer and anti-inflammatory effects (Table 1) [84,85]. Song and co-workers synthesized a series of 3-*O*-β-chacotriosyl ursolic acid serving as entry inhibitors of H5N1, and the SAR evaluation demonstrated a similar observation to their study of the derivatives of oleanolic acid. Amide modification at 28-COOH (**20**) and altering C-3 configuration from β to α (**21**) improved the selective index greatly [50,86,87,88]. Hong and co-workers synthesized several bis-disaccharide and mono-disaccaride-bearing dihydroxytriterpenes, and corosolic acid bis-lactoside was found to possess anti-α-glucosidase activity close to acarbose [51].

Sun and co-workers generated the analogues of calenduloside E (CE, Figure 4), isolated from *Aralia elata*, and evaluated their protective effects in preventing H9c2-cardiomyocytes apoptosis induced by oxidative stress. Compound **22** (Figure 4), 28-*N*-amido-3-*O*-β-_D_-galactopyranosyl ursolic acid, exhibited the best cytoprotective effect, and was found to inhibit ROS generation, maintain mitochondrial membrane potential and reduce caspase-dependent apoptosis cascade [52]. To elucidate the cytoprotective mechanism of CE, Sun’s group introduced an alkyne to 28-COOH of CE with amide linkage (**23**), and the binding with Hsp90AB1 was identified through the biotin-streptavidin system in human umbilical vein endothelial cells (HUVEC). Further biological assays demonstrated that CE can reverse the reduction in Hsp90AB1 after ox-LDL treatment and the binding of CE to Hsp90AB1 was increased in a dose-dependent manner [89]. On the other hand, the alkyne protein probe of the CE analogue (**24**) revealed a similar result when binding with Hsp90 [90].

### 2.3. Lupane

The most well-known synthetic lupane-type saponins are those derived from betulinic acid and lupeol, which are usually isolated from white birch bark [91]. Lupane-type saponins are reported to exhibit anti-inflammatory, antiviral and antitumor activities (Figure 4) [92,93,94]. However, the drug development of lupane-type compounds is limited by their low water solubility and poor pharmacokinetic profiles [95]. Efforts to improve these properties prior to 2011 have been completely reviewed [96]; therefore, our discussion was focused on the development of lupane-type saponins after 2011 (Table 1).

Legault and co-workers have demonstrated that a bidesmosidic betulinic saponin 28-*O*-α-_L_-rhamnopyranosyl betulin 3β-O-α-_L_-rhamnopyranoside (Bi-L-RhamBet) induced lung cancer cell death through the mitochondrial electron transfer chain which induced ROS production and decreased membrane potential, leading to apoptosis and DNA fragmentation [97]. Based on the findings, Pichette and co-workers synthesized α-_L_-rhamnose-containing betulinic and ursolic saponins, which exhibited cytotoxicity against human colorectal adenocarcinoma cells DLD-1 (IC_50_ = 5 μM) and low toxicity in human skin fibroblasts WS-1 (IC_50_ > 100 µM) in dirhamnoside-bearing betulinic derivatives (**25**) and anti-inflammatory activity was revealed in ursolic derivatives (**26**) [53]. Liberek and co-workers synthesized glucosamine- and galactosamine-containing betulinic saponins, revealing that galactosamine derivative (**27**) exhibited high cytotoxicity in both MCF-7 breast cancer cells and HDFa normal cells (1.7 and 4.2 µM, respectively), and had no inhibition activity toward bacteria and fungi [54]. Several peracetylated glucosamine-modified betulin and allobetulins have been synthesized by Kataev and co-workers, found to exhibit antibacterial efficacy (minimum inhibitory concentration, MIC = 7.8–15.5 µg/mL) against *Staphylococcus aureus* [56,57].

Pakulski and Strnad developed a series of lupane- (modified at 3-OH position) and homolupane-based (modified at 28-COOH position) saponins, including 3-O-glycoside, 28-COO-glycoside, 28-COO-thioglycoside, 28-COO-selenoglycoside and bidesmosidic saponins. However, all the compounds tested were either poorly cytotoxic to cancer cells, or poorly selective for cancer cells over normal cells [55,98,99,100,101].

### 2.4. Dammarane

Representative dammarane-type saponins are ginsenoside (isolated from *Panax ginseng*), gypenoside (from *Gymnostemma pentaphyllum*), bacoside (from *Bacopa monnieri*), and gymnemic acid (from *Gymnema sylvestre*), all of which have been extensively studied for their biological activities [5]. Although the naturally occurring dammarane-type saponins have been studied for a long time [102], SAR studies revealed that chemically generated synthetic saponins were mainly on ginsenoside derivatives and were still far from satisfactory [103,104]. Based on the recent progress in organic synthesis (Figure 4) [105,106], several studies showed cytotoxicity against cancer cells [58,107,108], as well as antioxidative [59], antiasthma [60] and skin-protective effects [61] of ginsenoside derivatives (Table 1).

Modifications to the C-20 side chain of the ginsenosides demonstrated that the hydroxylation site and configuration at C-20 and C-24 profoundly influenced cytotoxicity, in which ginsenoside derivatives bearing a monosaccharide at C-3 and a disaccharide at C-6 were more cytotoxic than those without modification [58,107,108]. Another study found that ginsenosides incorporating a *Z*-configuration at C-20 exhibited higher antioxidative activity than those not, and those bearing a disaccharide moiety at the C-3 position exerted a stronger hemolytic effect [59]. In other work, Li and co-workers evaluated the in vivo anti-IgE activity of a series of ginsenoside compound K (CK and **28**, Figure 4) analogues in an ovalbumin (OVA)-induced asthmatic mouse model; compounds modified at C20 and ketoxime exhibited a comparable antiasthmatic effect to CK [60]. Additionally, the conjugation of ginsenoside Rh2 onto polymer-functionalized zinc oxide nanocomposites generated water-dispersible materials capable of absorbing, scattering, and reflecting UV radiation, exemplifying their potential as nanomaterial sun-blockers [61].

## 3. Steroid Saponin

### 3.1. Cholestane

Studies of synthetic cholestane-type saponins have focused on SAR studies of OSW-1 and SBF-1 analogues, with monodesmosidic saponins isolated from *Ornithogalum saundersiae* in 1992 (Figure 5) [109], due to high antitumor efficacy and low toxicity (Table 1), and the SAR studies related to these structures were completely summarized in a review paper published in 2013 [110]. Recently, Lei reported several novel OSW-1 analogues bearing disaccharides consisting of cinnamoyl- or methoxybenzoyl-modified xylose and arabinose with either a (1 → 3) or (1 → 4) linkage at the 16-OH position. The conformation of the carbohydrate moiety was found to strongly influence antitumor activity, with the (1 → 3)-linked analogue exhibiting potent antiproliferative activity and adopting a chair conformation (**29**), whereas the (1 → 4)-linked analogue adopted a boat conformation and was inactive [62].

In 2019, Yu and co-workers developed a series of SBF-1 C22-ester analogues bearing 2-acylamine xylose residues which were up to 40-fold more cytotoxic to Jurkat and MDA-MB-231 cancer cell lines compared to SBF-1 and taxol, but they had no selectivity to normal cells. Although the cytotoxicity was lower, compounds with complex acyl group on 2-*N* position of xylose (**30**) were found to increase the selectivity index. Besides, an analogue with photoaffinity and clickable moiety was also synthesized and can be applied to target identification in the future [63].

Sakurai sought to understand the anticancer mechanisms of OSW-1 by conjugating a fluorescent moiety to the 3′’-position and studied the distribution of compounds in the cell by a fluorescence microscope. The probe was found to concentrate in the ER and Golgi within 10 min in a temperature-dependent manner [111]. OSW-1 was also derivatized selectively at the 4′’-position with a good to excellent yield using an organotin reagent, to give analogues bearing a fluorescent tag, biotin tag or alkyne moiety for further biological studies [112]. On the other hand, they also performed the 4′’ position-selective synthesis of a clickable photoaffinity probe, and confirmed its probing activity with a proof-of-concept experiment using bovine serum albumin [113]. Sequential derivatization was applied to the synthesis of dual functional OSW-1 derivatives **31** which bore fluorescence and photoaffinity moieties and had a comparable cytotoxicity to OSW-1 [114]. The deacetylated OSW-1 was known to exhibit a significantly lower cytotoxicity than the parent compound; however, Sakurai’s group synthesized **32** which displayed the same cell distribution pattern with fluorescence labeling, indicating that acetyl modification was not necessary for cell penetration [115]. A further mechanism study revealed that the ability of OSW-1 to trigger an apoptotic Golgi response was through CREB3-ARF4 pathway, and it might be an important reason in the selectivity between normal cells and cancer cells [64]. By accessing membrane-disrupting activity, the interaction between OSW-1 and cholesterol was found to be important in membrane permeabilization and hemolytic activity [65,116], but the membrane activity of OSW-1 proceeded without destroying membrane integrity comparing to other saponins, which might be another reason for the cytotoxic selectivity of OSW-1.

### 3.2. Spirostane

In the field of chemical synthesis, spirostane-type saponins are majorly derived from diosgenyl glycosides (Figure 5), widely distributed in legumes, fenugreek and yams, and are found to exhibit promising biological effects, such as metabolic regulation, anticancer, neuroprotective and antithrombotic activities (Table 1) [13,117,118,119,120]. In 2013, we reported the synthesis of a series of chlorogenin-type saponins with an orthogonal-protecting group strategy to modify the chacotriose moiety on the 3-OH of chlorogenin. Ten different glycoside donors, including di-, tri-, and tetrasaccharides, are synthesized and conjugated to 3-*O*-β-_D_-glucopyranosyl chlorogenin in 34–95% yields. Unfortunately, all of the derivatives show weak inhibition activity against CCRF, HL60, and PC-3 cancer cells and are inactive toward the inhibition of influenza virus entry [13]. Since attempts to modify dioscin failed, it was suggested that dioscin might have an unique property in dioscin-induced apoptosis in leukemic cells which was identified through the death receptor-mediated extrinsic and intrinsic apoptosis pathways. More importantly, dioscin induced a profound increase in protein expression of CCAAT/enhancer-binding protein α (C/EBPα), which was a critical factor for myeloid differentiation [66].

Recent studies focused on the antimicrobial and antitumor activities of spirostane-type saponins. For example, Liberek evaluated the antimicrobial and antifungal activities of diosgenyl 2-amino-2-deoxy-β-_D_-glucopyranoside, and its *N*-alkyl and *N*,*N*-dialkyl derivatives. The *N*-ethyl (**33**) and *N*-propyl derivatives were found to exhibit stronger activity against Gram-positive bacteria and *Aspergillus niger* than underivatized diosgenyl saponin [121]. The importance of the carbohydrate moiety was also investigated by conjugating the galactosamine and then synthesizing a series of *N*-acyl, 2-ureido and *N*-alkyl derivatives. Interestingly, the alteration of glucosamine into galactosamine greatly diminished the antibacterial and antifungal activity, but activities were recovered in the derivatives with *N*-ethyl, *N*-acetyl and 2-chloroethylureido modifications, which presented a similar trend to glucosamine derivatives [122]. Since the glucosamine-bearing diosgenyl saponins exhibited better activities, Liberek and co-workers replaced the *N*-alkyl into *N*-aminoacyl and *N*-hydroxyacyl moieties. *N*-aminoacyl saponins (e.g., **34**) exhibited ideal antifungal activity and growth inhibition effects in Gram-positive bacteria, and the compounds with better activity belonged to sarcosines or _L_- or _D_-alanines attached to glucosamine. The (*N*-acetyl)aminoacyl and *N*-hydroxyacyl analogues were all inactive, indicating the free amino group in glucosamine residue was necessary for their antifungal and antibacterial effects. With the exposure of the saponins to human erythrocytes, the hemolytic activities were found to be independent of the antibacterial efficacy [68].

Tigogenin is a spirostane-type sapogenin isolated from *Yucca gloriosa* L. and found to exhibit very low antiproliferative activity. However, tigogenin derivatives bearing saccharide chains showed greatly enhanced antitumor activities in several cancer cell lines, including HL-60 cells [123]. Zhang and co-workers synthesized a series of tigogenin saponins bearing different sugar moieties using an oxyamine neoglycosylation method. Tigogenin bearing a 2-deoxy-galactose (**35**) exhibited IC_50_ values of 2.7 and 4.6 µM against HepG2 and MCF7 cancer cell lines, respectively. On the other hand, the 3R-tigogenin neoglycosides showed enhanced antitumor activity and the 3S isomers were not active, suggesting that the configuration of glycosidic bonds was important for cytotoxicity against cancer cells [67].

### 3.3. Furostane

Furostane-type saponins are characterized by a hemiketal ring and carbohydrate moiety attached to the 3-OH and/or 26-OH of sapogenin. Easy conversion of furostane-saponins to spirostane-saponins and the microheterogeneity made the isolation of furostane-saponins from natural sources become problematic and difficult. Therefore, several chemical works have been established for the synthesis of furostane-type saponins using 26-hydroxy-16, 22-dioxo-cholestan as a building block, providing efficient tools to expand SAR knowledge for medicinal chemistry research [124,125,126,127].

Li and co-workers developed a concise and practical synthetic route for the synthesis of several furostane-type saponins from readily accessible 16β-acetoxy-22-oxo-cholestanic derivatives, which can be applied to SAR studies. The biological assay results demonstrated that funlioside B is a promising lead for α-glucosidase inhibition (Table 1) [69].

### 3.4. Cardenolide

The earliest known cardiac glycosides (CGs) were isolated from *Digitalis purpurea* L. and have served as an oral treatment for heart failure and cardiac arrhythmia for centuries [128]. The cardioprotective activity of CG originates from their binding and inhibition of the cardiac myocyte membrane Na^+^/K^+^-ATPase, lowering the heart burden [129]. Interestingly, a recent drug-repurposing study found that cardiac glycosides triggered cancer cell apoptosis through various mechanisms [130], including exposure of calreticulin on the outer leaflet of cell membranes, active secretion of ATP via autophagy, and release of nonhistone chromatin-binding protein high mobility group box 1 (HMGB1) [131,132,133]. Several cardiac glycosides are now under investigation in Phase I and II clinical trials for solid tumor treatments and revealed a promising safety and efficacy [134].

The potent apoptotic effect (Table 1) of the cardiac glycosides spurred studies of their SARs (Figure 5). Tang and co-workers focused on the derivatization of the 3-*O* sugar moiety of digoxigenin to understand its influence on Nur77 protein induction, cytotoxicity, and apoptosis induction in NIH-H460 cancer cells. Applying oxyamine neoglycosylation to construct a series of 6-deoxy and 2,6-dideoxy-_D_-glucose derivatives of digoxin revealed that the induction of Nur77 protein expression was strong in some compounds but lacked sufficient cytotoxicity against cancer cells [135]. To further improve the cytotoxicity against NIH-H460 cancer cells, digoxigenin derivatives containing different carbohydrates at 3-OH were synthesized, and the *O*-linked saponins bearing _D_-ribose and _L_-rhamnose (**38**, **39**) exhibited nanomolar range cytotoxicity which was better than digoxin. However, the inductions of the Nur77 protein only showed a low correlation to cytotoxic effects [70]. The biotinylated cardiac glycoside probes were derived from α-antiarin and β-antiarin. They were found to increase Nur77 protein inductions. However, significant decrements of cytotoxicity against NIH-H460 cells were observed in all derivatives, indicating that the probe can only be applied to the study of Nur77 protein induction rather than cytotoxic efficacy in cancer cells of cardiac glycosides [136].

Total syntheses of sarmentogenin, trewianin, ouabagenin and other cardenolides have also been reported. Inoue and co-workers synthesized the sapogenin moiety of sarmentogenin and trewianin, as well as their glycosylated derivatives, revealing the important role of 3-*O*-glycosylation for cytotoxicity against MCF-7 cells [71]. Nagorny used a Cu (II)-catalyzed diastereoselective Michael/aldol cascade approach to synthesize cardiac glycosides incorporating carbon atoms in various states of oxidation. _L_-α-rhamnosides of cannogenol (**40**), strophanthidol (**41**), and digitoxigenin (**42**) were found to exhibit broad cytotoxicity against cancer cells and were nontoxic at 3 µM against normal cells and a developing fish embryo. The anticancer activity of these compounds involved DNA-damage upregulation and apoptosis induction [72]. Kinghorn and co-workers synthesized several (+)-digoxin derivatives. The SAR studies stated that C12 and C14 hydroxyl groups and an unsaturated lactone moiety were important for cytotoxicity; the 3-O-glycosyl moiety seemed to be less crucial [137]. Collectively, these reports have expanded the chemical tools available for the synthesis of cardiac glycosides and their derivatives.

## 4. Discussion

Saponins are a large and diverse class of natural products with intriguing and varied biological activities, but their development is limited by the microheterogeneity and difficulties associated with their isolation. Chemical synthesis has now been established as a powerful tool for the derivatization of saponins and the study of their SAR, leading to numerous exciting discoveries of molecules more potent and less toxic than those originally isolated [6]. The roles of carbohydrate in the bioactivity of saponin has been discussed extensively in cytotoxicity assays and can significantly influence the activity. In assays against cancer cells, the existence of carbohydrates showed both positive [58,71,107,108] and negative [37,55,98,99,100,101] effects. Changing the composition of sugar chains could enhance the cytotoxic effect [14,29,67,70] but also showed no differences in some studies [13,34]. Nevertheless, the sugar chains containing monosaccharides were usually better than di- and polysaccharides [14,29]. These results showed the importance of a carbohydrate moiety in influencing biological activity.

The recently reported biological assays of synthetic saponin were focused on several disease models, including anticancer, anti-infection (antiviral, antibacterial, antileishmanial), immunomodulatory (immunostimulatory, antioxidant, anti-inflammation), antiglucosidase, and cytoprotective effects (anti-Alzheimer, antihemolytic, sun protection) (Table 1, Figure 6). The anticancer activity is the most adopted biological assay in saponins and evaluated in most of the scaffolds except quillaic acid- and ursane-type saponins. Oleanane-saponins exhibited antitumor cytotoxicity in the range of IC_50_ = 0.5–7.6 µM against various type of cancer cells in vitro, including cervical cancer (SMMC-7721, SGC-7901), lung cancer (NCI-H460, A549), colorectal cancer (HCT-116, HT-29), astrocytoma (U251), ovary cancer (SKOV-3), breast cancer (MCF-7), prostate cancer (PC-3), liver cancer (HepG2), leukemia (HL-60), and lymphoma (U937). The in vivo mice study revealed the tumor regression activity in H22 xenograft and MCF-7 xenograft models. The proposed anticancer cytotoxicity pathways in oleanane-saponin involved mitochondrial-dependent intrinsic apoptosis [31,35], apoptosis-ferroptosis-M2-TAM polarization [30], PI3K/Akt/mTOR pathway [21], and G1 cell cycle arrest [45]. Lupane-saponins were found to exhibit in vitro antitumor cytotoxicity in IC_50_ = 0.9–5.0 µM in colorectal cancer (DLD-1), breast cancer (MCF-7), leukemia (CEM), cervical cancer (HeLa), and melanoma (G-361); >100 and 1.3 µM in normal WS-1 and BJ cells, respectively. Cancer cell death was reported to be related to induced ROS production and decreased mitochondrial membrane potential, leading to apoptosis and DNA fragmentation [97]. Dammarane-saponins were found to exhibit anti-HeLa cytotoxicity with IC_50_ = 4.6 µM. Cholestane-saponins exhibited antitumor cytotoxicity in the broad range of IC_50_ = 0.0012–13.0 µM in colorectal cancer (HCT-116), lung cancer (NCI-H1975), pancreatic cancer (Capan2, SW1990), neuroblastoma (SK-N-SH), cervical cancer (BGC823), liver cancer (HepG2), acute T cell leukemia (Jurkat, CCRF), and 0.16 µM in normal CRL1999 cells. One of the anticancer cytotoxicity was related to apoptotic Golgi responses through the CREB3-ARF4 pathway [64]. Spirostane-saponins were active toward HepG2 and MCF-7 with IC_50_ = 2.7 and 4.6 µM, respectively. Cardenolide-saponins showed nanomolar range IC_50_ in lung cancer (NCI-H460), breast cancer (MCF-7, MDA-MB-231), cervical cancer (HeLa), glioblastoma (U87), and liver cancer (HepG2), and no toxicity in normal HEK293T and NIH-3T3 cells at 3 µM. The cytotoxicity of cardenolide-saponin was found to involve DNA-damage upregulation, causing the induction process of apoptosis [72].

Anti-infection activities were found in oleanane-, hederagenin-, ursane-, lupane-, and spirostane-type saponins. Oleanane-saponins exhibited anti-H5N1 influenza virus activity with IC_50_ = 4.05 μM and low toxicity to Madin-Darby canine kidney (MDCK) cells (92.7 μM) [39,40]. As for antileishmanial assay, oleanane-saponins exhibited ED_50_ against axenic L. *mexicana* amastigotes as 6.0 and 29.7 μM to the macrophage, which was not selective enough and needed further optimization [46]. In ursane-saponins, anti-H5N1 influenza virus activity was found, exhibiting IC_50_ = 4.05 µM accompanied with a low toxicity to MDCK cells (>950 µM) [50]. Lupane-saponins were effective in inhibiting the growth of *Staphylococcus aureus*, and the MIC value was 7.8 μg/mL [56,57]. Spirostane-saponins also showed anti-infection activity against *Candida*, *Staphylococcus*, *Enterococcus*, and *Bacillus* species in the range of 2–8 µM/mL [68].

The saponins with immunomodulatory effects were categorized into upregulation and downregulation. Immune upregulation activities were majorly evaluated in quillaic-saponins, which were extensively studied for the enhancement of serum IgG production compared to GPI-0100 and QS-21 for the development of vaccine adjuvants [47,48,49]. As for immune downregulation activities, ursane-saponins were able to protect H9c2 cardiomyocytes from H_2_O_2_-induced apoptosis in a dose-dependent manner (0.02–0.5 µg/mL) through the inhibition of ROS generation and maintenance of mitochondrial membrane potential which were possibly related to the binding of Hsp90AB1 [52,89], and prevented NO overproduction induced by lipopolysaccharides (LPSs) in macrophages with EC_50_ = 9.8 µM [53]; dammarane-saponins were found to be effective in reducing serum IgE levels and airway resistance in an OVA-induced asthmatic mouse model [60]. The activity of glucosidase inhibition was evaluated in oleanane-, ursane-, and furostane-type saponins. Oleanane-, ursane-, furostane-saponins were able to inhibit α-glucosidase with IC_50_ = 9.2, 448, 96 µM, respectively, and the positive control, acarbose, was found to inhibit α-glucosidase in the 500–1000 µM range [41,51,69]. Cytoprotective effects were found in oleanane-saponins for the protection of neuron cells from H_2_O_2_- and Aβ_1–42_-induced injury [38]; dammarane-saponins for the protection of HaCaT cells from sunlight by increased sun protection factor (SPF) rating [61] and prevention of 2, 2′-azobis(2-methylpropionamidine) dihydrochloride (AAPH)-induced hemolysis in rabbit erythrocytes [59].

## 5. Conclusions

The biological and pharmacological activities of synthetic saponins in recent studies were collected and categorized with the saponins’ structural characteristics. More detailed studies employing state-of-the-art biological experiments are necessary to understand the detailed mechanism of saponins in multiple disease models. By summarizing the SAR studies of synthetic saponins, we expect to help scientists working in this field to have a deeper understanding of the relationship between the structure and biological activity of synthetic saponins and hope to provide fruitful information for the development of saponin-based bioactive compounds.

## Figures and Tables

**Figure 1 molecules-25-04974-f001:**
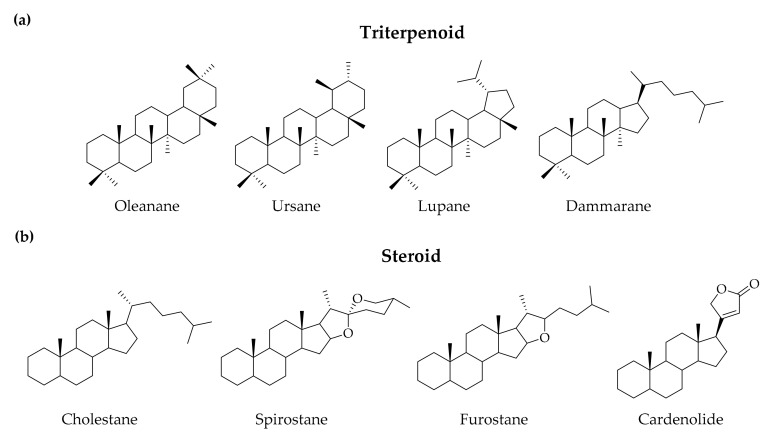
Representative sapogenin structure of (**a**) triterpenoid saponins and (**b**) steroid saponins.

**Figure 2 molecules-25-04974-f002:**
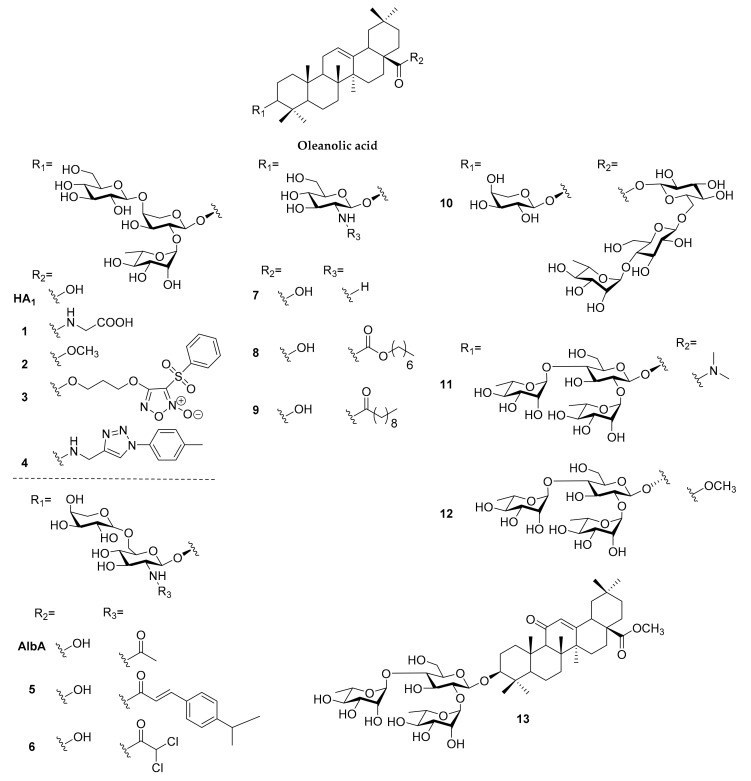
Synthetic saponins derived from oleanolic acid.

**Figure 3 molecules-25-04974-f003:**
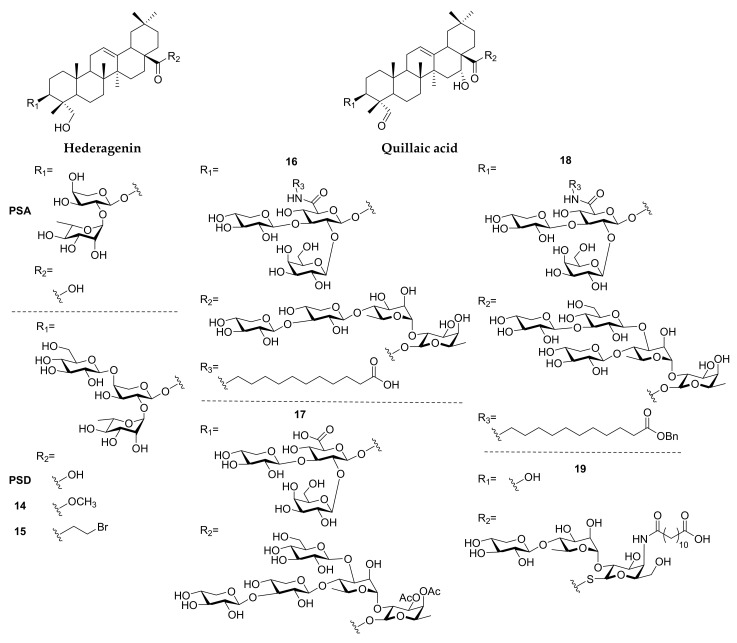
Synthetic saponins derived from hederagenin and quillaic acid.

**Figure 4 molecules-25-04974-f004:**
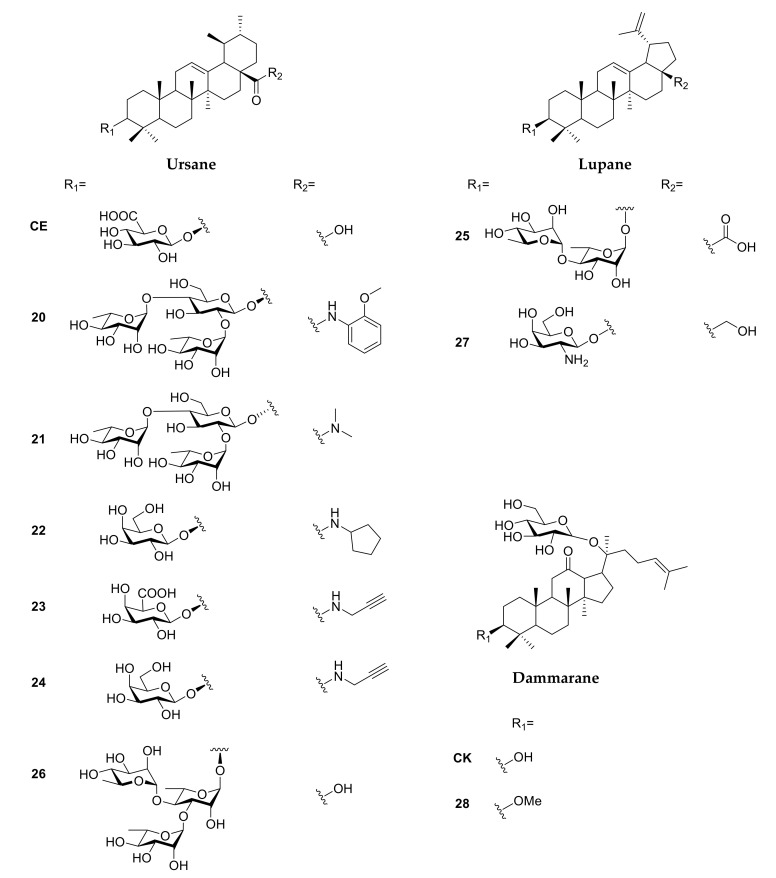
Synthetic saponins derived from ursane-, lupane-, and dammarane-type sapogenins.

**Figure 5 molecules-25-04974-f005:**
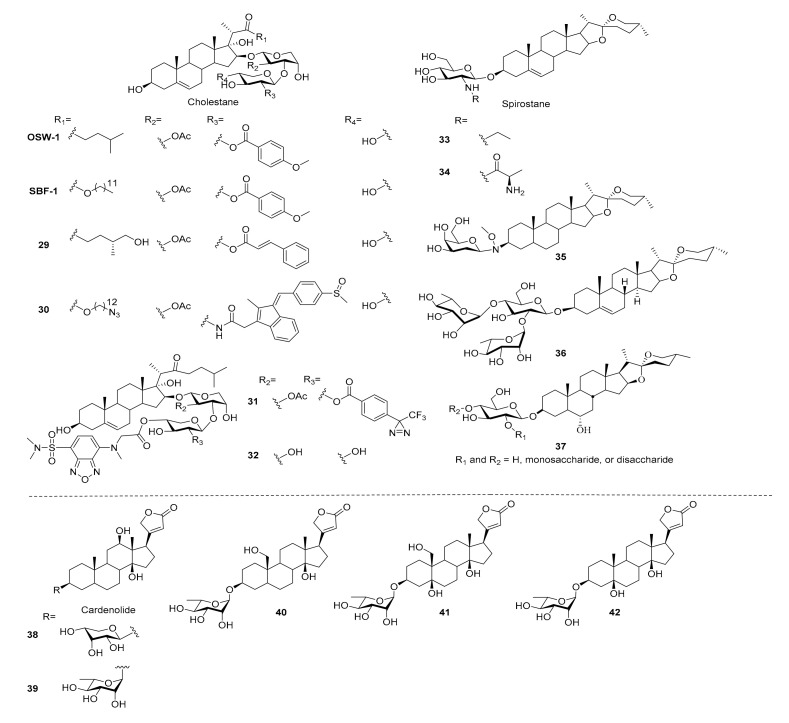
Synthetic saponins derived from steroid-type sapogenins.

**Figure 6 molecules-25-04974-f006:**
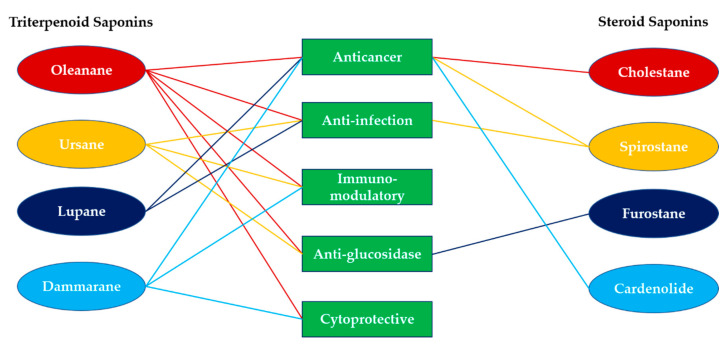
Summary of the biological and pharmacological activities of synthetic saponins.

**Table 1 molecules-25-04974-t001:** Biological effects of the synthetic saponins.

Sapogenin	Pharmacology	Effect	Reference
Triterpenoid Saponins			
Oleanane (Oleanolic acid)	Anticancer	IC_50_ = 1.1–4.6 µM against SMMC-7721, NCI-H460, U251, SKOV-3, HCT-116, SGC-7901 in vitro and 46.8% regression in H22 xenograft mouse	[33]
		IC_50_ = 1.6–6.5 µM against SMMC-7221, NCI-H460, U251, HCT-116 in vitro and 51.5% regression in H22 xenograft mouse	[32]
		IC_50_ = 0.5–2.7 µM against PC-3, HT-29, HepG2, A549, HL-60, U937 in vitro	[35]
		IC_50_ = 7.6 µM against HCT-116 in vitro and >50 µM to human normal cells	[31]
		50% regression in MCF-7 xenograft mouse	[30]
		IC_50_ = 0.76 µM against HL-60	[18]
		IC_50_ = 5.74 and 2.78 μM against HL-60 and HCT-116, respectively	[21]
	Anti-Alzheimer	Protec PC12 from H_2_O_2_ and Aβ_1–42_ induced injury	[38]
	Antivirus	Prevent H5N1 infection in MDCK as the selective index > 40	[39,40]
	Glucosidase inhibition	Inhibit α-glucosidase with IC_50_ = 9.2 µM which is 40X stronger than acarbose	[41]
Oleanane (Hederagenin)	Anticancer	IC_50_ = 1.2–4.7 µM against SMMC-7721, MCF-7, NCI-H460, A549, HCT-116 in vitro and 49.8% regression in H22 xenograft mouse	[45]
	Antileishmanial	ED_50_ = 6.0 µM against axenic L. *mexicana* amastigotes and 29.7 μM against host macrophage	[46]
Oleanane (Quillaic acid)	Immunomodulatory	IgG expression higher than GPI-0100	[47,48]
		IgG expression higher than QS-21	[49]
Ursane	Antivirus	Prevent H5N1 infection in MDCK as the selective index > 950	[50]
	Glucosidase inhibition	Inhibit α-glucosidase with IC_50_ = 448 µM which is comparable to acarbose	[51]
	Antioxidant	Protect H9c2 cardiomyocytes from H_2_O_2_ induced apoptosis in a dose-dependent manner (0.02–0.5 µg/mL)	[52]
	Anti-inflammatory	EC_50_ = 9.8 µM for preventing NO overproduction induced by LPS in macrophages	[53]
Lupane	Anticancer	IC_50_ = 5.0 µM and >100 against DLD-1 and WS-1, respectively (high selectivity index)	
		IC_50_ = 1.7 and 4.2 µM against MCF-7 and HDFa, respectively (low selectivity index)	[54]
		IC_50_ = 0.9–2.6 µM against CEM, MCF-7, HeLa, G-361 and 1.3 μM against BJ normal cells	[55]
	Antibacterial	MIC = 7.8–15.5 µg/mL against Staphylococcus aureus	[56,57]
Dammarane	Anticancer	IC_50_ =4.6 µM against HeLa cells	[58]
	Antioxidant	Inhibit the AAPH-induced hemolysis in rabbit erythrocytes	[59]
	Antiasthma	Effective in reducing IgE plasma level and airway resistance in OVA-induced asthmatic mouse model	[60]
	Skin protection	Conjugation of ginsenoside Rh2 with ZnONcs can increase SPF rating and reduce toxicity in HaCaT cells	[61]
**Steroid Saponins**			
Cholestane	Anticancer	IC_50_ = 0.0012–13.0 µM against HCT-116, NCI-H1975, Capan2, SW1990, SK-N-SH, BGC823, HepG2.	[62]
		IC_50_ = 0.0054 and 0.16 µM against Jurkat cancer cells and CRL1999 normal cells, respectively	[63]
		Identify the apoptosis induced by OSW-1 might result from Golgi response through CREB3-ARF4 pathway and interaction between OSW-1 and cholesterol	[64,65]
Spirostane	Anticancer	IC_50_ = 1–2 µM against leukemia cell CCRF and induced a profound increase in protein expression of CCAAT/enhancer-binding protein α (C/EBPα)	[13,66]
		IC_50_ = 2.7 and 4.6 µM against HepG2 and MCF-7, respectiviely	[67]
	Antimicrobial and Antifungal	MIC = 2–8 µg/mL against *Candida*, *Staphylococcus*, *Enterococcus*, *Bacillus* species	[68]
Furostane	Glucosidase inhibition	Inhibit α-glucosidase with IC_50_ = 96 µM which is 12X stronger than acarbose	[69]
Cardenolide	Anticancer	Inhibit 70% of NIH-H460 cancer cells at 50 nM	[70]
		IC_50_ = 0.108–3.27 µM against MCF-7	[71]
		Inhibit cancer growth in 0.01–0.1 µM range and no toxicity at 3 µM	[72]

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
