# Peer review of "Biological and Pharmacological Effects of Synthetic Saponins"

_molecules, 2020, doi:10.3390/molecules25214974_

Round 1

Reviewer 1 Report

This is an interesting review in which the studies related to the synthesis of different saponin models are analyzed in order to study the relationship between biological activity and structure. The review can be a reference text for all those researchers interested in the development of new bioactive molecules based on saponins. It refers to the main groups of both triterpenic and steroidal saponins, the article is well structured and very informative.

Author Response

Thank you very much for the positive feedbacks. We will continue to dedicate in this field.

Reviewer 2 Report

The manuscript provides a review on synthetic saponins, their structure and biological and pharmacological effects. Overall, it is well written and organized, addressing the chemical synthesis of these compounds, and comparing their properties to those of the ones originally isolated from natural sources. Nevertheless, I would suggest a minor modification for further publication. Table 1, which provides information on the biological effects of synthetic saponins, is not sited in the text, only in the conclusions. I would suggest the authors to refer the table along the manuscript since it contains information concerning the distinct saponins discussed. Moreover, according to the title, this is an important part of the article, reflecting the biological and pharmacological effects of the studied compounds, which justifies its discussion in the different sections of the article.

Author Response

Thank you for the positive feedbacks and suggestions. We refer the Table 1 throughout the manuscript to inform readers if they need a complete summary of biological activities. As per suggestion from the reviewer, we incorporated Table 1 to the text. Furthermore, we construct a discussion section from line 383 to 453 and a new figure (Figure 6) to completely summarize the biological and pharmacological effects of saponins mentioned in this review.

Reviewer 3 Report

The proposed article reviews the literature on both structural and pharmacological data on natural and synthetic saponin derivatives. This review is well structured and help understanding the structure-activity relationships for different saponin subfamilies. Figures are adequately chosen to illustrate the incredible structural features of saponins. One could suggest to the authors to add one additional figure illustrating the cellular pharmacological targets, presenting then a possible biological mechanism.

Author Response

Thank you for the positive feedbacks and suggestions. We add a figure (Figure 6) and a discussion section in line 383 to 453 to summarize the biological and pharmacological effects of saponins mentioned in the review. The detailed cellular target and mechanism are not included in the figure to prevent the possible misleading effect due to the incomplete result of saponin mechanism. Instead, we describe the related pathway in discussion section for the readers who are interested in studying specific saponin scaffold or biological effect.

Reviewer 4 Report

Bibliography should be carefully checked. The Authors do not have examined the bibliography on this topic.

Author Response

We feel sorry if there are some missing pieces in the field of chemically synthetic saponins, and we have tried our best to collect the literatures mainly in recent five years. Several descriptions in the manuscript are modified to be clearer that we only focus on the aspect of synthetic saponin derivatives. The English writing is proof read by a native English speaker in this field and have been extensively checked again. Thank you.

Round 2

Reviewer 4 Report

Manuscript can be accepted.